# Navigating the Modern Landscape of Sepsis: Advances in Diagnosis and Treatment

**DOI:** 10.3390/ijms25137396

**Published:** 2024-07-05

**Authors:** Jin Ho Jang, Eunjeong Choi, Taehwa Kim, Hye Ju Yeo, Doosoo Jeon, Yun Seong Kim, Woo Hyun Cho

**Affiliations:** 1Division of Pulmonary, Allergy and Critical Care Medicine, Department of Internal Medicine, Transplantation Research Center, Research Institute for Convergence of Biomedical Science and Technology, Pusan National University Yangsan Hospital, Yangsan 50612, Republic of Korea; jjhteen1@naver.com (J.H.J.); graceejchoi@gmail.com (E.C.); taehwagongju@naver.com (T.K.); dugpwn@naver.com (H.J.Y.); sooli10@hanmail.net (D.J.); yskim@pusan.ac.kr (Y.S.K.); 2Department of Internal Medicine, School of Medicine, Pusan National University, Yangsan 50612, Republic of Korea

**Keywords:** sepsis, septic shock, biomarker, diagnosis, treatment, immunomodulation, novel

## Abstract

Sepsis poses a significant threat to human health due to its high morbidity and mortality rates worldwide. Traditional diagnostic methods for identifying sepsis or its causative organisms are time-consuming and contribute to a high mortality rate. Biomarkers have been developed to overcome these limitations and are currently used for sepsis diagnosis, prognosis prediction, and treatment response assessment. Over the past few decades, more than 250 biomarkers have been identified, a few of which have been used in clinical decision-making. Consistent with the limitations of diagnosing sepsis, there is currently no specific treatment for sepsis. Currently, the general treatment for sepsis is conservative and includes timely antibiotic use and hemodynamic support. When planning sepsis-specific treatment, it is important to select the most suitable patient, considering the heterogeneous nature of sepsis. This comprehensive review summarizes current and evolving biomarkers and therapeutic approaches for sepsis.

## 1. Introduction

Sepsis is defined as a life-threatening organ dysfunction caused by a dysregulated host response to infection, according to the Third International Consensus Definitions for Sepsis and Septic Shock (Sepsis-3). Septic shock, a subset of sepsis, is defined when vasopressors are required to maintain blood pressure despite adequate volume resuscitation and a lactic acid level > 2 [1]. Sepsis and septic shock have become more common over time; by 2017, there were 48.9 million cases and 11 million annual deaths from sepsis-related causes [2,3]. Each year, 270,000 sepsis-related deaths and 1.7 million sepsis cases are recorded in the US [4]. As sepsis continues to threaten global health, advancements in its diagnosis and treatment are required to achieve improved clinical outcomes.

Over the past few decades, the definition of sepsis has changed several times [5]. Sepis-3, the most recent study published in 2016, defines sepsis as organ failure resulting from a dysregulated host response to an infectious disease [1]. Given the importance of early recognition, it is challenging to define sepsis. First, confirming infection remains a rate-limiting step in the early diagnosis of sepsis. Conventional culture methods have the disadvantage of a long turnaround time to detect and specify microorganisms. In addition, 3–50% of sepsis cases are reported as culture-negative [6,7]. Secondly, the sequential organ failure assessment (SOFA) score has been used as an index to predict organ failure, but there is a limitation that it is not specific for sepsis or infection [8]. Additionally, requiring laboratory values in the pre-hospital stage is a clear limitation of the SOFA score. To overcome this, the quick SOFA (qSOFA) score was proposed, but unlike the results of the initial validation study, the qSOFA score has shown low sensitivity in diagnosing sepsis [9,10]. The National Early Warning Score (NEWS), a diagnostic tool comprised of six physiological parameters, has gained attention for demonstrating higher sensitivity in diagnosing sepsis compared to qSOFA [11,12,13]. However, NEWS detects general clinical deterioration and is based on a range of physiological parameters, thus lacking specificity in diagnosing sepsis [14,15]. With the development of our understanding of the pathophysiology of sepsis, biomarkers have emerged to circumvent the limitations of current diagnostic approaches. Biomarkers for sepsis can help in the early diagnosis, prognosis prediction, treatment monitoring, and stratification of patients for individualized treatment. Biomarkers, such as soluble receptors, membrane receptors, damage-associated molecular patterns (DAMPs), cytokines, chemokines, acute phase proteins, and non-coding RNAs, are currently being studied for the diagnosis of sepsis and the prediction of prognosis [16,17].

Despite new sepsis definitions and cumulative epidemiological evidence regarding the benefits of early detection, specific treatments for sepsis are still lacking. The Surviving Sepsis Campaign (SSC) guidelines mostly comprise the best supportive management of hemodynamic support and preventive care against sepsis-associated complications [18]. However, this classical therapeutic approach is limited because it does not adequately address the inflammatory response activated and amplified by the immune system stimulated by infectious molecules, which is the main mechanism of sepsis. Therefore, as the pathophysiology of sepsis continues to be discovered, immunomodulation has recently emerged as a promising adjuvant therapy. These therapeutic methods can be broadly divided into pharmacological and extracorporeal immunomodulation.

In this review, novel biomarkers of sepsis are introduced and discussed, and the upcoming evidence for new technologies for immunomodulation therapies is discussed.

## 2. Novel Biomarkers for Sepsis

The overall mortality from sepsis has decreased over the past decades owing to the introduction of the SSC guidelines [19,20,21]. Nonetheless, sepsis is a major cause of death worldwide, accounting for approximately 20% of all global deaths in 2017 [2]. If treatment is initiated in the early phases of sepsis that have not progressed to organ failure, the mortality rate may decrease [22]. For each hour of delay in the start of antibiotic treatment, the mortality rate increases by 7 to 12% [23,24]. Unfortunately, with the revision of the sepsis definition to facilitate early detection, there is still a lack of awareness regarding sepsis diagnosis. Biomarkers have been identified that overcome these limitations and enable the initiation of appropriate treatment at an early stage of sepsis. To date, more than 250 biomarkers of sepsis have been studied and applied in various clinical settings [17,25,26]. In this review, we focus on biomarkers related to the diagnosis of sepsis and classify them according to their pathophysiology (Table 1). Subsequently, the biomarkers capable of distinguishing sepsis from systemic inflammatory response syndrome (SIRS) are summarized in Table 2.

### 2.1. Damage-Associated Molecular Patterns

#### 2.1.1. Calprotectin

The cytoplasm of neutrophils contains high concentrations of calprotectin, a heterodimeric calcium- and zinc-binding protein made up of S100A8 (calgranulin A) and S100A9 (calgranulin B) subunits [54,55]. When neutrophils are activated by infectious molecules, calprotectin is secreted into circulation. Subsequently, it binds to receptors, including toll-like receptor 4 (TLR4) and advanced glycation end products, to increase the production of pro-inflammatory cytokines and amplify the inflammatory cascade [26]. The blood concentration of calprotectin increases rapidly within a few hours of exposure to bacteria or endotoxins; hence, it may be helpful in the diagnosis of sepsis [56]. The value of calprotectin in the diagnosis of sepsis has been confirmed in several studies [57,58,59]. Another study showed that procalcitonin (PCT) (area under the curve [AUC], 0.736; 95% confidence interval [CI], 0.625–0.829) was inferior to calprotectin (AUC, 0.775; 95% CI, 0.667–0.861) in discriminating between bacterial and viral pneumonia [60]. In a meta-analysis assessing the usefulness of calprotectin in sepsis diagnosis, the following values were discovered: 0.77, 0.85, 5.20, and 0.27 for the pooled sensitivity, specificity, positive likelihood ratio, and diagnostic odds ratio, respectively [31].

#### 2.1.2. High Mobility Group Box 1 (HMGB1)

High mobility group box 1 (HMGB1) is a short protein with a 185 amino acid region made up of two tandem high-mobility group (HMG) boxes, A box and B box, and a stretch of 30 amino acid residues near the carboxy-terminus that is made up of glutamic and aspartic acid residues [61]. To preserve the stability of the nucleosome and control transcription, translation, and DNA repair, HMGB1 binds to DNA intracellularly [62,63]. HMGB1 can be actively released from cells due to external stressors or passively released due to cell death (e.g., apoptosis, pyroptosis, and necroptosis) [64]. Extracellular HMGB1 acts as a DAMP and induces a proinflammatory state after binding to immunomodulators, such as ribonucleic acid (RNA), histones, and lipopolysaccharides (LPS) [65]. In a mouse model of sepsis, HMGB1 was detectable in circulation 8 h after the onset of sepsis and peaked at 32 h [66]. In the same study, the delayed administration of an HMGB1 antibody inhibited endotoxin-induced lethality. A study involving patients with sepsis and septic shock demonstrated that HMGB1 is a modulator of the late inflammatory response, as its level persists for approximately a week following hospitalization [67]. When diagnosing sepsis-associated acute kidney injury, measuring HMGB1 using a combination of blood and urine samples showed a better application value than measuring it alone (sensitivity 88%, specificity 87%, accuracy 88%, and AUC 0.891) [32]. In another study, plasma HMGB1 provided a clue for the diagnosis of sepsis (sensitivity of 75.8%, specificity of 41.3%, and AUC of 0.684) and showed that its combination with human β-defensin 3 would be useful in the diagnosis of sepsis [33].

### 2.2. Membrane Receptors

#### CD64

The immunoglobulin Fc gamma receptor I with high affinity, CD64, is constitutively expressed in monocytes, eosinophils, and macrophages [68]. Additionally, CD64 expression in neutrophils is low in healthy individuals; however, when infected with bacterial pathogens, the expression of CD64 increases more than 10-fold within a few hours [69]. In a systematic review and meta-analysis, including 1986 patients from eight studies, neutrophil CD64 (nCD64) had a pooled sensitivity and specificity of 0.76 and 0.85 for the diagnosis of sepsis [34]. A more recent meta-analysis compared the accuracy of sepsis diagnosis using nCD64, PCT, and C-reactive protein (CRP) across 14 trials involving 2471 patients [35]. In this study, nCD64 had a bigger area under the summary receiver operating characteristics curve than either PCT (0.89 [95% CI, 0.84–0.95] vs. 0.84 [95% CI, 0.79–0.89]; *p* < 0.05) or CRP (0.89 [95% CI, 0.87–0.92] vs. 0.84 [95% CI, 0.80–0.88]; *p* < 0.05) [35]. The measurement of nCD64 seems to be helpful in the diagnosis of sepsis, but it has limitations, such as manual sample preparation or a long incubation period due to the use of flow cytometry. To overcome these limitations, a study was conducted to measure the bedside expression of nCD64 using a smartphone-based microfluidic biochip. An excellent linear correlation exists between flow cytometry and the utilization of smartphone-imaged microfluidic chips, as indicated by a correlation coefficient R^2^ = 0.82 (slope = 0.99) [70].

### 2.3. Soluble Receptors

#### 2.3.1. Presepsin (Soluble CD14 Subtype)

The glycoprotein CD14 is expressed on the surfaces of immune cells, including macrophages and monocytes, and it is an LPS receptor that belongs to the class of toll-like receptors. When the pro-inflammatory cascade is triggered by infection, the N-terminus of CD14, including presepsin, is cleaved and enters the bloodstream [71]. Phagocytosis and lysosomal cleavage of microbes are the physiological functions of presepsin. Because presepsin is relatively specific for bacterial infection and its concentration increases in the early stages of sepsis, it can be used as a biomarker for the early detection of sepsis [72]. Several meta-analyses have demonstrated the importance of presepsin for the accurate diagnosis of sepsis [36,37,38]. Previous studies have shown that presepsin is not inferior to other biomarkers for diagnosing sepsis. Based on a meta-analysis encompassing 19 observational studies with 3012 patients, the pooled sensitivity and specificity for the diagnosis of sepsis were 0.84 (95% CI, 0.80–0.88) and 0.73 (95% CI, 0.61–0.82) for presepsin and 0.80 (95% CI, 0.75–0.84) and 0.75 (95% CI, 0.67–0.81) for PCT. In this study, the AUCs of presepsin and PCT were 0.87 and 0.84, showing relatively similar diagnostic performance [29]. While the biological mechanism of presepsin is linked to the body’s reaction to Gram-negative bacterial infections, presepsin levels may not discriminate between Gram-positive and Gram-negative bacterial infections. Although most previous studies have reported higher mean or median presepsin levels in Gram-negative sepsis than in Gram-positive sepsis, only three studies have shown statistically significant differences [50,73,74,75,76,77]. According to a previous in vitro study, Gram-positive bacteria, such as *Staphylococcus aureus*, may trigger presepsin production at a level similar to that of Gram-negative bacteria [78]. Therefore, it can be assumed that the induction of presepsin expression is due to the unique immunogenicity of individual pathogens rather than endotoxins [79].

#### 2.3.2. Soluble Triggering Receptor Expressed on Myeloid Cell-1 (sTREM-1)

Triggering receptor expressed on myeloid cells-1 (TREM-1), a member of the immunoglobulin superfamily, is expressed on neutrophils, monocytes, and monocytes [80]. When TREM-1 interacts with LPS or other ligands, the inflammatory response is amplified, resulting in an uncontrolled immune response and organ failure [81,82]. Soluble TREM-1 (sTREM-1) is a soluble form of TREM-1 released from the cell surface. sTREM-1 can be observed in various body fluids, including the plasma, pleural fluid, and urine, when infectious diseases are present; therefore, it can be used to differentiate between infectious and non-infectious causes in the diagnosis of sepsis [83,84,85]. Compared to conventional biomarkers for infectious diseases, such as PCT and CRP, sTREM-1 is a more sensitive and specific biomarker [85]. The capacity of sTREM-1 to identify sepsis had a pooled sensitivity and specificity of 0.82 (95% CI, 0.73–0.89) and 0.81 (95% CI, 0.75–0.86) and an AUC of 0.88 (95% CI, 0.85–0.91) in a meta-analysis of 19 trials involving 2418 patients [39]. Furthermore, in another meta-analysis, sTREM-1 demonstrated a high sensitivity of 0.85 (95% CI, 0.76–0.91) and moderate specificity of 0.79 (95% CI, 0.70–0.86) in discriminating sepsis from SIRS [40]. However, studies have reported that CRP and interleukin-6 (IL-6) predict sepsis and septic shock more accurately than sTREM-1 [86,87,88]. According to a study by Jedynak et al., there is no difference in plasma sTREM-1 levels between noninfectious SIRS and sepsis [88]. According to other studies, non-infectious diseases also show an increase in sTREM-1 levels [89,90]. However, since the definition of sepsis was different in each of the above studies and the sample size was not large, additional prospective studies are needed to verify the diagnostic accuracy of sTREM-1 levels.

#### 2.3.3. Soluble Urokinase-Type Plasminogen Activator Receptor (suPAR)

Various cell types, such as neutrophils, lymphocytes, macrophages, monocytes, and other immune cells, express the urokinase-type plasminogen activator receptor (uPAR) [91]. Proteases cleave uPAR from the cell surface in response to inflammatory activation, resulting in a soluble form of the receptor that is observed in various body fluids, such as plasma and urine [91,92,93]. A 1995 report showed that 13 patients with sepsis in the intensive care unit (ICU) had high levels of soluble urokinase-type plasminogen activator receptor (suPAR) in their plasma. The plasma level of suPAR is high in various diseases exhibiting severe inflammatory reactions, such as infectious, autoimmune, and neoplastic diseases [53,94,95,96]. In 2016, a meta-analysis of nine studies with 1237 patients revealed that the overall AUC of suPAR was 0.82 for the diagnosis of bacterial infection. Additionally, suPAR’s pooled sensitivity and specificity for identifying infections were 0.73 and 0.79, respectively [41]. A recent meta-analysis of 30 studies involving 6906 patients reported that the diagnostic accuracy of suPAR was comparable to that of PCT for sepsis. In relation to sepsis diagnosis, the pooled sensitivity and specificity of suPAR were 0.76 (95% CI, 0.63–0.86; *p* < 0.01) and 0.78 (95% CI, 0.72–0.83; *p* < 0.01). The AUC for suPAR in sepsis was 0.83 (95% CI, 0.80–0.86), suggesting a moderate level of diagnostic accuracy. Subgroup analysis revealed that the AUC of suPAR to differentiate between SIRS and sepsis was 0.81 (95% CI, 0.77–0.84), with corresponding sensitivity and specificity of 0.67 (95% CI, 0.58–0.76) and 0.82 (95% CI, 0.73–0.88) [42]. To address its relatively low sensitivity, combining suPAR with other biomarkers, rather than using it alone, may help to increase the diagnostic accuracy. In this regard, a study on the combined use of HMGB1 and suPAR for the diagnosis and prognosis of sepsis in acute respiratory distress syndrome was conducted [97].

### 2.4. Cytokine and Chemokine

#### 2.4.1. Interleukin-6 (IL-6)

IL-6, a product of T cells discovered by Hirano et al. in the 1980s, activates B cells and promotes antibody production [98]. IL-6 is involved in the acute phase of inflammation, and various studies have addressed its potential use as a biomarker of sepsis [27,28,99,100]. According to a 2015 meta-analysis, IL-6 has a sensitivity of 80.0%, a specificity of 75.0%, and an AUC of 0.868 for detecting early sepsis [101]. In a meta-analysis of 22 studies, including 2680 patients, published the following year, the pooled sensitivity and specificity of IL-6 for the diagnosis of sepsis were 0.68 and 0.73, respectively, and the AUC for differentiating sepsis from SIRS was 0.80. Additionally, IL-6 showed a diagnostic value comparable to PCT, but due to its relatively low sensitivity, it was recommended as a diagnostic tool to confirm infection rather than rule out infection in patients with SIRS [27]. Variations exist in the outcomes of studies comparing the diagnostic accuracy of IL-6 and other biomarkers for sepsis. In a study conducted on 142 patients with sepsis and septic shock diagnosed according to Sepsis-3, IL-6 was able to distinguish sepsis from the control group (AUC, 0.89; 95% CI, 0.83–0.94; *p* < 0.001; sensitivity, 80.4%; specificity, 88.9%) and septic shock from sepsis (AUC, 0.80; 95% CI, 0.71–0.89; *p* < 0.001; sensitivity, 76.1%; specificity, 78.4%). For the diagnosis and prognosis of sepsis and septic shock, IL-6 was more useful than PCT and pentraxin 3 (PTX3) [30]. A more recently published meta-analysis showed that IL-6 had a lower diagnostic value compared to CD64 and PCT (AUC, 0.77; 95% CI, 0.73–0.80; sensitivity, 72.0%; specificity, 70.0%) [28]. Polymorphic variation in the promoter region of the IL-6 gene, depending on race, increases the risk of sepsis; therefore, it seems necessary to consider race in future studies comparing IL-6 with other biomarkers [102].

#### 2.4.2. Monocyte Chemoattractant Protein-1 (MCP-1)

Monocyte chemoattractant protein-1 (MCP-1) is also known as C-C motif chemokine ligand 2 (CCL2) [103]. MCP-1 is produced in a variety of cells, including monocytes and endothelial cells, and is activated by growth factors, cytokines, and oxidative stress [103,104]. MCP-1 mediates monocyte migration and immune cell recruitment to damaged sites [105,106]. Numerous studies have demonstrated the significance of MCP-1 in the pathophysiology of sepsis [107,108,109]. In one observational study, serum MCP-1 levels were significantly higher in the first days of septic shock and provided a significant and valuable AUC for differentiating septic shock patients from healthy and postoperative controls [110]. In another study that included trauma patients, day 1 plasma levels of MCP-1 were a major risk factor for the development of sepsis. Additionally, the AUC of MCP-1 in predicting sepsis in trauma patients was observed to be 0.82 (95% CI, 0.71–0.93; *p* < 0.093; [111]. In a recently published meta-analysis, the combined AUC was 0.90 (95% CI, 0.87–0.92), indicating a moderate level of accuracy in sepsis diagnosis, and the combined sensitivity and specificity were 0.84 (95% CI, 0.70–0.92) and 0.82 (95% CI, 0.67–0.91), respectively [43].

### 2.5. Acute Phase Protein

#### 2.5.1. Pentraxin 3 (PTX3)

Pentraxin 3 (PTX3), an acute-phase protein, is a member of the pentraxin superfamily [112,113]. Moreover, it is produced in various cells, such as endothelial cells, monocytes, smooth muscle cells, fibroblasts, adipocytes, and dendritic cells, in response to different stimuli, such as LPS, tumor necrosis factor-α (TNF-α), interleukin-1 (IL-1), and TLR agonists [112,113,114,115,116]. PTX3 is a major factor in the human innate immune system that plays an important role in activating the classical complement pathway and regulating inflammation [115,117]. Increased PTX3 levels have been repeatedly reported to be associated with sepsis and organ damage [118,119,120]. In a clinical study conducted on 213 patients with sepsis and septic shock defined according to Sepsis-3, PTX3 levels were consistently and significantly higher during the observation period than in the control group (*p* < 0.001). Additionally, PTX3 levels on days 1, 3, and 8 were able to significantly differentiate sepsis from septic shock (range of AUC 0.73–0.92, *p* = 0.0001) [121]. In a prospective study published in 2019, the AUC of PTX3 for differentiating sepsis from the control group was observed to be 0.84 (95% CI, 0.95–0.99; *p* < 0.001), and the sensitivity and specificity were 92.6% and 97.4%, respectively, at a cut-off value of 15.10 ng/mL. However, the diagnostic value of IL-6 was superior to that of other biomarkers, including PTX3 [30]. Therefore, it is necessary to determine whether the value of PTX3 in the diagnosis of sepsis is valid in multicenter studies or meta-analyses.

#### 2.5.2. Adrenomedullin (ADM) and Mid-Regional Pro-Adrenomedullin (MR-proADM)

Adrenomedullin (ADM) is a 52-amino acid peptide hormone produced by various cell types, including endothelial and vascular smooth cells [122]. Its primary biological actions include positive inotropic, vasodilatory, natriuretic, diuretic, and bronchodilatory effects [123]. Because circulating ADM is quickly broken down and removed from the blood, quantification using conventional immunoassay techniques is challenging [26]. The mid-regional fragment of pro-adrenomedullin (MR-proADM) consists of amino acids 45–92 and is separated from the final proADM molecule at a ratio of 1:1 with ADM [124]. MR-proADM has been investigated as a sepsis biomarker because it is more stable than the ADM peptide and therefore easier to measure [125,126]. In a prospective observational study conducted on 120 patients, the initial MR-proADM levels in individuals with SIRS and organ dysfunction aided in determining the cause of infection. Predictive diagnostic power was shown with an AUC of 0.9474, and the sensitivity and specificity were 0.80 (95% CI, 70.8–87) and 0.9375 (95% CI, 62.5–100), respectively [124]. In other studies, MR-proADM showed diagnostic value in differentiating between septic and non-septic origins in patients with SIRS [127,128]. In a recent meta-analysis of 11 studies, including 2038 cases, MR-proADM had high sensitivity and specificity of 0.83 (95% CI, 0.79–0.87) and 0.90 (95% CI, 0.83–0.94), respectively, for diagnosing sepsis. Additionally, the best cut-off value of MR-proADM for diagnosing sepsis was 1–1.5 nmol/L, and the AUC was 0.91 [129]. In a more recent meta-analysis, including 40 studies, presepsin and MR-proADM showed good diagnostic performance, with the AUC being 0.90 for presepsin and 0.91 for MR-proADM [44]. A double monoclonal sandwich immunoassay has been developed to measure C-terminally amidated physiologically active ADM (bio-ADM) [130,131]. In a prospective study of 215 patients, the septic shock group had higher levels of bio-ADM than the sepsis group (110.3 vs. 45.3 pg/mL, *p* < 0.001) [132]. Similarly, in the AdrenOSS-1 study, the level of bio-ADM was significantly higher in the septic shock group than in the sepsis group [133]. Because there is a gap in the function and clearance kinetics of MR-proADM and ADM, bio-ADM will need to be used in future large-scale studies.

### 2.6. Angiogenic Growth Factors

#### Angiopoietin

The angiopoietin/Tie2 system, together with its downstream signaling pathways, plays a significant role in the regulation of vascular maturation, stability, and integrity during angiogenesis [134]. This system has been reported to be associated with endothelial cell damage and vascular dysfunction in sepsis [135,136]. Ang1 and Ang2 bind to the angiopoietin receptor (Tie2) on endothelial cells; Ang1 is a Tie2 agonist, and Ang2 is an antagonist of Tie2. In the vasculature, Ang1 protects against vascular leakage and maintains endothelial quiescence, whereas Ang2 breaks down the endothelial intracellular junctions and promotes increased vascular permeability [137,138,139]. In several clinical studies on sepsis, both high Ang2 and low Ang1 levels or high Ang2/Ang1 or low Ang1/Ang2 ratios were associated with poor prognosis and organ failure [140,141]. In a prospective observational study of 105 patients, Ang2 levels increased as the severity of sepsis progressed. In addition, Ang2 levels showed excellent diagnostic performance for distinguishing the sepsis group from the control group (AUC = 0.97). Ang2 had better diagnostic performance in distinguishing septic shock from sepsis than Ang1 or the Ang1/Ang2 ratio (AUC = 0.778) [45]. In another study, plasma Ang2 levels in patients with sepsis and septic shock were significantly higher than those in healthy controls (*p* < 0.05). Additionally, the AUROC value of the Ang2 level for differentiating sepsis and septic shock was 0.631 (95% CI, 0.464–0.799; *p* = 0.1288), which was higher than that of the Ang1 level (95% CI, 0.320–0.683; *p* = 0.9904) [110]. Plasma Ang2 levels may be an additional biomarker of sepsis-associated coagulopathy. In comparison with healthy controls, patients with sepsis and suspected disseminated intravascular coagulation (DIC) had higher Ang2 levels, according to a study involving 102 patients. Since Ang2 levels are markedly elevated in sepsis-associated coagulopathy, patients with sepsis may be risk-stratified into non-overt and overt DIC using this biomarker [142]. Angiopoietin is a promising biomarker for the diagnosis of sepsis, the prediction of prognosis, and the development of novel therapeutic strategies.

### 2.7. Non-Coding RNAs

#### 2.7.1. MicroRNA (miRNA)

The microRNA (miRNA) is one of many small non-coding RNAs (ncRNAs). A non-coding RNA is an RNA molecule that is transcribed from DNA but is not translated into proteins. The biological roles of miRNAs include RNA silencing and the regulation of post-translational gene expression [17]. Since miRNAs bind to RNA-binding proteins or are transported in the form of exosomes or microvesicles, they are less likely to be degraded in the environment and can be easily measured by methods, such as polymerase chain reaction or microarrays; therefore, they have been studied as biomarkers for various diseases, including sepsis [26,143]. Evidence exists that miRNAs play major roles in mediating the host response to infection, primarily by regulating proteins involved in the innate and adaptive immune systems [144,145]. An ROC curve analysis demonstrated the ability of miRNA-125a (AUC, 0.749; 95% CI, 0.695–0.803) and miRNA-125b (AUC, 0.839; 95% CI, 0.795–0.882) to distinguish sepsis from healthy controls in a study including 150 patients. Meanwhile, only miRNA-125b was effective for treatment and prognostication in patients with sepsis [146]. A recent meta-analysis of 2337 patients, including 14 patients with SIRS, 2 local infection patients, and 14 healthy controls, provided information on the diagnostic accuracy of miRNAs. When identifying sepsis, the sensitivity and specificity of miRNA were 0.80 (95% CI, 0.75–0.83) and 0.85 (95% CI, 0.80–0.89), respectively, and the AUC was 0.89 (95% CI, 0.86–0.92), showing that it is a highly accurate method [46]. However, a limitation of existing studies is that the relationship between miRNAs and diseases cannot be revealed in detail because of the heterogeneity that exists between studies.

#### 2.7.2. Long Non-Coding RNAs (lncRNAs)

Long non-coding RNAs (lncRNAs) are non-coding RNAs that are composed of transcripts of more than 200 nucleotides that are not translated into proteins [147]. Several studies have linked various lncRNAs to both innate and adaptive immune responses [148]. When comparing patients with sepsis to healthy controls, the expression of long non-coding RNA-nuclear enriched abundant transcript 1 (lnc-NEAT1) was higher and the expression of miRNA-124 was lower in patients with sepsis. These differences allow for the identification of patients with sepsis from healthy controls [149]. For non-small cell lung cancer, lncRNA metastasis-associated lung adenocarcinoma transcript 1 (MALAT1), sometimes referred to as lnc-NEAT2, is a prognostic marker [150]. Moreover, it regulates the expression of pro-inflammatory cytokines triggered by LPS, including IL-6 and TNF-α, by blocking NF-κB activity [151]. Liu et al. reported that the lnc-MALAT1/miRNA-125a axis was elevated in patients with sepsis relative to healthy controls (*p* < 0.001) and had an excellent AUC of 0.931 (95% CI, 0.908–0.954) in differentiating patients with sepsis from healthy controls [152]. Another prospective cohort study of 120 patients with sepsis demonstrated that lnc-MALAT1 was superior to the Acute Physiology and Chronic Health Assessment (APACHE) II score (AUC 0.868) and lactate levels (AUC 0.868) in its ability to reliably identify sepsis (AUC 0.910) and predict 28-day survival (AUC 0.886) [153]. Long non-coding RNA maternally expressed gene 3 (lnc-MEG3), one of the other lncRNAs, has been linked to increased inflammation and organ damage [154,155]. Compared with healthy controls, patients with sepsis had lower levels of miRNA-21 expression and higher levels of lnc-MEG3 expression and the lnc-MEG3/miRNA-21 axis. The AUCs for lnc-MEG3 (0.887; 95% CI, 0.856–0.917) and the lnc-MEG3/miRNA-21 axis (0.934; 95% CI, 0.909–0.958) were shown to be useful in predicting an elevated sepsis risk, whereas the AUC for miRNA-21 (0.801; 95% CI, 0.758–0.844) demonstrated a good predictive value for a decreased sepsis risk [156]. Because we do not yet know much about the detailed functions or mechanisms of ncRNAs, additional research is needed to understand the pathophysiology of ncRNAs in sepsis.

### 2.8. Changes in Plasma Biomarker Concentrations during Sepsis

The changes in biomarker concentrations within the first 72 h of the onset of sepsis are illustrated in Figure 1. Procalcitonin exhibits a rapid increase, peaking sharply at around 6 h, followed by a gradual decline over the subsequent hours [157]. Similarly, IL-6 shows a swift rise, reaching its peak between 6 to 8 h [158]. The HMGB1 concentration increases more gradually, reaching a plateau at around 12 to 24 h and maintaining elevated levels before starting to decline towards the 72 h mark [159]. Presepsin levels rise quickly, peaking at approximately 6 h, and then decline gradually over the remaining period [160]. sTREM-1 follows a similar pattern, with a rapid rise peaking at around 6 h and a subsequent gradual decrease [161]. suPAR shows a steady increase, peaking at around 24 h, followed by a gradual decline over the next 48 h [162]. The CD64 concentration rises sharply, peaking at about 6 h, and then maintains higher levels before gradually decreasing [163]. MCP-1 levels peak at around 8 h, with a sharp initial rise followed by a steady decline over the remaining hours [103]. The PTX3 concentration exhibits a rapid increase, peaking at around 8 h, and then gradually declines over time [164]. MR-proADM shows a steady increase, peaking at around 24 h, followed by a gradual decrease towards the end of the observation period [164]. Finally, calprotectin demonstrates a rapid increase, peaking sharply at around 6 h, followed by a gradual decline [164]. Each biomarker follows a distinct pattern, highlighting the importance of monitoring multiple biomarkers to understand sepsis progression and severity.

## 3. Novel Immunomodulatory Therapies in Sepsis

Hospital mortality due to sepsis has decreased with the development of treatments, such as the introduction of SSC guidelines. However, the quality of life after survival has often decreased, and patients may not lead a long-term life due to an uncontrolled immune response and resulting organ failure [18,165,166]. Approximately half of patients with sepsis recover, one-third die within a year, and one-sixth exhibit severe, long-lasting disabilities [167]. This may be a limitation of conventional therapeutic approaches, such as antibiotics and organ support, which apply non-individualized methods to heterogeneous syndromes, such as sepsis. Moreover, due to the long-term consequences of sepsis involving physical and cognitive impairment, the need for a comprehensive therapeutic approach has been emphasized [166,168]. Accordingly, immunomodulatory therapies have begun to receive attention for improving the survival of patients with sepsis, where both excessive inflammation and immunosuppression coexist. This review summarizes the latest updates in immunomodulatory therapies, including blood purification (Table 3).

### 3.1. Targeting the Hyperinflammation of Sepsis

#### 3.1.1. Interleukin-1 (IL-1)

Interleukin-1 (IL-1) plays a key role in the hyperinflammatory phase of sepsis, along with TNF-α. When IL-1α or IL-1β binds to the IL-1 receptor, it activates downstream signaling [195]. While the use of an IL-1 receptor antagonist (IL-1 RA) has shown promising results in in vitro and animal studies, randomized controlled trials (RCTs) in humans have not demonstrated a survival benefit [169,196,197]. In a phase 3 study with 893 patients with sepsis, IL-1 RA did not significantly improve the 28-day survival compared to the control group. However, patients with severe organ failure or a predicted death risk greater than 24% showed improved survival [170]. A subsequent phase 3 study was terminated early due to the lack of significant survival benefits between the IL-1 RA (33% mortality) and control groups (36% mortality) in the first interim analysis [171]. A post hoc analysis of a previous study indicated that IL-1 RA improved survival in sepsis with hepatobiliary dysfunction and/or DIC (65% vs. 35% in the placebo group) [198]. Another retrospective study found that a recombinant human IL-1 receptor antagonist (rhIL-1RA) significantly reduced mortality in patients with baseline plasma IL-1 RA levels of 2071 pg/mL or higher [199]. Recently, IL-1 RA has been studied in viral sepsis, such as COVID-19. In patients with an suPAR level of 6 or higher, anakinra treatment improved the 28-day mortality [200]. Further research is needed to confirm the efficacy and safety of anakinra through biomarker stratification in patients with sepsis caused by non-viral pathogens.

#### 3.1.2. Interleukin-6 (IL-6)

During infection and tissue damage, IL-6 is rapidly and transiently released into the bloodstream, enhancing host defense by inducing acute-phase hematopoiesis and immune responses [201]. In severe infections like sepsis, targeting the inflammatory cascade amplified by IL-6 has been a major focus for treatment development [202,203]. However, the benefits of IL-6 blockade in sepsis are controversial. The use of an IL-6 monoclonal antibody did not show a survival benefit in the mouse endotoxic shock model induced by LPS or TNF-α [204]. Conversely, another study reported beneficial effects of IL-6 blockade in a rodent sepsis model [205]. Recently, the inhibition of both membrane and soluble IL-6 receptors using monoclonal antibodies, such as tocilizumab or sarilumab, has been successfully applied to critically ill COVID-19 patients [172,206,207]. Additionally, a recent meta-analysis of 11,643 patients showed that IL-6 blockade was associated with reduced mortality, even in non-COVID-19 sepsis [208]. Although IL-6 is known for its pro-inflammatory effects, it can also act as an anti-inflammatory or protective molecule in various contexts [209]. Thus, changes in the IL-6 concentration or activity represent a double-edged sword. Identifying the underlying causes of these changes may help to improve the prognosis of sepsis.

#### 3.1.3. The Complement System

The complement system, activated by pathogen invasion, tissue damage, and DAMPs, plays a crucial role in defending against invaders [210,211]. However, in sepsis, its activation can cause tissue damage and organ failure [212]. Studies on primates, such as baboons, have shown that targeting complement activation can prevent coagulation defects and organ failure [213,214,215]. Among them, C5a and its receptors, C5aR and C5aR2, have shown promise as therapeutic targets in sepsis [173,216]. A phase II RCT investigated the effects of vilobelimab, a monoclonal anti-C5a antibody, in early sepsis or septic shock. Vilobelimab selectively neutralizes C5a without inhibiting the membrane attack complex or causing safety concerns. Additionally, higher doses led to longer ICU- and ventilator-free days for patients [217]. C3a is also significant in platelet aggregation during sepsis, contributing to DIC and organ failure [218,219,220,221]. Compstatin, which inhibits the cleavage of C3, has been shown to reduce fibrinogen and platelet consumption and renal damage in a baboon model of *Escherichia coli* sepsis [222]. However, recent findings indicate that platelet aggregation induced by *E. coli* depends on C3b rather than C3a, necessitating further research [223].

#### 3.1.4. Adrenomedullin

Sepsis induces endothelial cell dysfunction, leading to vasodilation, edema, hypotension, and subsequent organ failure [23]. Among the molecules involved in these mechanisms, bio-ADM has attracted considerable attention as a therapeutic target. While intravascular bio-ADM protects against capillary leakage, extravascular bio-ADM increases permeability and induces vascular smooth muscle relaxation, leading to shock [168]. Adrecizumab, a non-neutralizing monoclonal anti-ADM antibody, has shown promise in improving endothelial dysfunction, reducing organ failure, and lowering mortality in preclinical sepsis models [174,224]. In the phase 2 RCT (AdrenOSS-2), involving 301 patients with sepsis, adrecizumab was found to be safe and effective [175]. In addition, a post hoc analysis revealed that patients with low levels of circulating dipeptidyl peptidase 3 (<50 ng/mL) experienced a more significant survival benefit, demonstrating the potential for biomarker-driven individualized treatment with adrecizumab [225].

#### 3.1.5. TREM-1

As previously mentioned, LPS increases the expression of TREM-1 in monocytes and neutrophils. The activation of TREM-1 in immune cells mediates dysregulated immune responses in sepsis by triggering the secretion of pro-inflammatory cytokines and chemokines [80,226]. Nangibotide, a 12-amino-acid polypeptide derived from TREM-like transcript-1 (TLT-1), inhibits activation of the TREM-1 pathway in sepsis by binding to the TREM-1 agonist ligand. In phase 1 trials, nangibotide was safe and well-tolerated at dosages up to 6 mg/kg/h for 7 h and 45 min, followed by a 15 min loading dose of up to 5 mg/kg [176]. A follow-up phase 2a trial involving patients with septic shock showed significant improvements in organ function (SOFA score), especially in those with high sTREM-1 levels [177]. The phase 2b trial conducted in 2019 included 42 hospitals across seven countries, evaluating the safety and efficacy of nangibotide at doses of 0.3 mg/kg/h or 1.0 mg/kg/h. While the primary endpoint was not achieved at an sTREM-1 value of ≥400 pg/mL, exploratory analyses with a cut-off of 532 pg/mL showed that high-dose nangibotide significantly improved the SOFA score from baseline to day 5 compared to the placebo [227]. However, no survival benefit was observed, and mortality was higher in the low-dose group.

#### 3.1.6. Extracorporeal Blood Purification

Extracorporeal blood purification (EBP) is an adjunctive therapy for modulating dysregulated immune responses in sepsis. These treatments attenuate immune responses by eliminating pathogen-associated molecular patterns, DAMPs, and inflammatory cytokines [228,229]. Different approaches have been employed to remove the mediators produced during sepsis, including hemofiltration, hemoadsorption, hemoperfusion, intermittent or continuous high-volume hemofiltration, plasmapheresis, and coupled plasma filtration and adsorption [230]. In this study, we discuss several EBPs widely used in clinical practice.

The polymyxin B (PMX)-immobilized fiber column (Toraymyxin^®^; Toray, Tokyo, Japan) is among the most frequently used endotoxin removal devices. PMX is a cationic peptide antibiotic with a high affinity for endotoxins through ionic and hydrophobic bonds [231]. In a hemoperfusion column cartridge, PMX is attached and immobilized on polystyrene fibers to enable endotoxin clearance without deleterious systemic effects [232]. Based on clinical studies, this approach is widely used in Japan; however, RCTs conducted in Western countries have reported unfavorable results regarding its efficacy [179,233]. A multicenter RCT on patients who underwent emergency surgery for sepsis caused by peritonitis found increased mortality in the PMX group compared to the control group, with no improvement in organ failure [180]. However, this ABDOMIX study has limitations, as it was conducted in low-risk populations with a low mortality rate of less than 20% in the control group and a median SOFA score of 10 points in both groups. The EUPHRATES study, the largest to date, conducted in the US and Canada, compared the PMX group with a group combining sham hemoperfusion and standard treatment in patients with septic shock and endotoxemia, defined as an endotoxin activity assay (EAA)  ≥ 0.60. No difference was observed in the 28-day mortality between the two groups in the overall subjects and the subgroup with multiple organ dysfunction scores exceeding nine points [181]. However, EAA cannot precisely quantify circulating endotoxins when the levels are above 0.9, and values within this range may not respond to treatment. Therefore, a post hoc analysis of the EUPHRATES study was performed on patients with EAA between 0.6 and 0.89, linking PMX to a 10.7% absolute mortality benefit at 28 days compared to sham patients (OR, 0.52; 95% CI, 0.27–0.99; *p* = 0.047) [234]. To date, several meta-analyses of PMX treatment have been published, with inconsistent conclusions [235,236,237,238,239]. Chang et al. found a pooled risk ratio of overall mortality for PMX therapy of 0.81 (95% CI, 0.70–0.95; *p* = 0.007). Furthermore, a noteworthy decrease in mortality was noted in intermediate- and high-risk groups, but not in the low-risk group, when patients were stratified according to the mortality rate in the conventional treatment group [235]. However, a more recent meta-analysis revealed contrasting results [239]. The survival benefit of PMX treatment was more pronounced in groups with Acute Physiology and Chronic Health Assessment II scores < 25 and sepsis than in those with severe sepsis or septic shock. The contrasting results are due to differences in the included studies and inconsistent definitions of disease severity. Future studies should clarify inclusion and exclusion criteria and standardize the disease severity evaluation.

CytoSorb^®^ is a cartridge composed of polystyrene and divinylbenzene microbeads with a porous and biocompatible polyvinylpyrrolidone cover (CytoSorbents, Monmouth Junction, NJ, USA) capable of removing hydrophobic pro- and anti-inflammatory mediators with molecular weights ranging from 5 to 60 kD. Moreover, both in vivo and in vitro studies have shown that it can remove pathogen-associated molecular patterns, DAMPs, complement factors, growth factors, myoglobin, bilirubin, and bile acids [240,241,242]. The efficacy of CytoSorb is concentration-dependent, suggesting that its removal efficiency increases with high water concentrations [230]. Several case series and observational studies have shown survival benefits and hemodynamic improvements without significant side effects [182,243,244]. However, the results from RCTs have been dismal. In an RCT of 97 patients, CytoSorb treatment did not show a significant reduction in IL-6 levels or mortality compared to the control group. In the unadjusted analysis, the treatment group had a higher mortality rate than the control group (44.7% vs. 26.0%, *p* = 0.039) [183]. Similarly, recent meta-analyses have reported either no survival benefit [245,246] or even higher mortality in the treatment group [247]. However, interpretation requires caution due to study heterogeneity, inconsistencies in adsorbent use, variations in the treatment duration, differences in time from diagnosis to first treatment, and blood flow rates. Future studies should address these inconsistencies and stratify groups to optimize the effectiveness of CytoSorb treatment.

oXiris^®^ (Baxter, Meyzieu, France) is a hemofilter that improves the adsorptive properties of the AN69ST membrane, coated with polyethyleneimine and pre-grafted with heparin. The AN69 core membrane has a high adsorptive affinity for cytokines, the polyethyleneimine layer adsorbs negatively charged endotoxins, and heparin lowers local thrombogenicity [248,249]. Therefore, oXiris is the only hemofilter that simultaneously performs renal replacement therapy, endotoxin removal, and cytokine adsorption. An in vitro study comparing three types of blood purification therapies showed that oXiris had a similar removal rate of inflammatory mediators to CytoSorb and a similar endotoxin removal rate to Toraymyxin [241]. A case series on oXiris in sepsis-associated acute kidney injury (S-AKI) showed a significant reduction in the SOFA score after 48 h of use, although no significant differences were observed in the length of ICU stay or in-hospital mortality [184]. In addition, a randomized double-blind crossover study of 16 patients with S-AKI who had endotoxin levels greater than 0.03 EU/mL demonstrated better efficacy in eliminating endotoxin and cytokines using oXiris compared to the standard AN69ST hemofilter [185]. The most recent meta-analysis of 14 studies, including 695 patients with sepsis, found that using oXiris reduced the 28-day mortality (OR, 0.53; 95% CI, 0.36–0.77, *p* = 0.001) and ICU stay compared to other filters. Additionally, the SOFA score, norepinephrine dose, IL-6 and lactate levels, and 7- and 14-day mortality were significantly lower in the oXiris group; however, no differences were observed in 90-day mortality, ICU mortality, or hospital mortality [250]. Nevertheless, due to the primarily observational design of the original study and the unknown risk of bias in the RCTs, the effectiveness of oXiris filters remains uncertain. Although the use of oXiris hemofilters in patients with sepsis is increasing, they have not been established as a standard treatment due to a lack of high-quality research and clinical guidelines. Future trials, including the Global ARRT International Registry (NCT03807414), should validate the efficacy of oXiris, its usage frequency, and patient suitability.

#### 3.1.7. Limitations of Therapies Targeting the Hyperinflammation of Sepsis

Immunosuppressive therapies in sepsis face significant limitations, including the heterogeneity of sepsis, which complicates the identification and targeting of immune pathways. The optimal timing for these therapies is uncertain, with risks of ineffectiveness or harm if administered too early or late. Additionally, accurately identifying patients who would benefit the most from these interventions remains challenging. These therapies also increase the risk of secondary infections by dampening the immune response in already vulnerable patients [251,252]. The immune response in sepsis is complex, involving both pro-inflammatory and anti-inflammatory processes, making balanced modulation difficult. Furthermore, the lack of reliable biomarkers to guide therapy and monitor effectiveness complicates treatment. Many therapies also lack robust clinical trial data, leaving their efficacy and safety uncertain.

Improving treatment for hyperinflammation in sepsis involves several strategies. Personalized medicine can enhance outcomes through the use of biomarkers and genetic profiling to tailor therapies to individual patients [253]. Early intervention and precise dosing can reduce inflammation without causing immunosuppression, thus lowering the risk of secondary infections. Combining anti-inflammatory agents with antibiotics and supportive care can address multiple aspects of sepsis, while developing new drugs and using biologic agents can more precisely modulate the immune response. Focusing on immune modulation rather than suppression can restore the immune balance, and adaptive therapies can be adjusted based on the patient response. Large-scale clinical trials and real-world evidence are essential for validating new treatments. The continuous monitoring of patient responses and incorporating patient feedback can improve the quality of life and long-term outcomes. Training healthcare providers and developing standardized protocols will ensure consistent and effective sepsis management. Addressing these areas can significantly improve treatment outcomes and reduce mortality in patients with sepsis.

### 3.2. Targeting Sepsis-Induced Immunosuppression

#### 3.2.1. Interferon-γ (IFN-γ)

In sepsis, monocytes are deactivated, and the expression of human leukocyte antigen (HLA)-DR and inflammatory cytokines decreases [254]. Candidate molecules were searched under the assumption that modulating the hypoinflammatory state of sepsis would improve its prognosis. IFN-*γ* is secreted from T_H_1 cells and activates macrophages, natural killer (NK) cells, and neutrophils, enhancing their antigen presentation and phagocytic ability [255]. The beneficial effects of IFN-*γ* treatment on sepsis were first demonstrated in an open-label study published in 1997. In this study, eight of nine patients with sepsis treated with IFN-γ survived, and two of them who discontinued treatment relapsed. Additionally, IFN-γ restored TNF-α production and HLA-DR expression in monocytes in a dose-dependent manner [256]. IFN-*γ* treatment was successful in subsequent studies in patients with sepsis and reduced HLA-DR expression in monocytes [257,258,259]. An RCT on the use of IFN-*γ* in sepsis-induced immunosuppression was conducted but was terminated early due to slow enrollment. This is because the expression of HLA-DR in CD14-monocytes is less than 30%, which is the diagnostic standard for immunoparalysis and is inappropriate [186]. Another RCT is in progress using the modified diagnostic criteria (less than 5000 HLA-DR receptors on CD14 monocytes) for immunoparalysis obtained in this study [260]. Case reports showed that immune recovery through IFN-γ injection was effective even for fungal sepsis [261,262]. However, Kim et al. reported conflicting results, suggesting that high levels of IFN-γ in the early stages of sepsis increase the risk of secondary *Candida* infection. In this study, increased IFN-γ inhibited the phagocytic activity of macrophages, which is essential for the clearance of pathogens. Through a transcriptomic analysis, they revealed that NK cells regulated by invariant natural killer T cells produce IFN-γ through the mTOR pathway during endotoxemia. The inhibition of mTOR by rapamycin during sepsis reduced IFN-γ secretion by NK cells, normalized the phagocytic function of macrophages, and improved the survival of secondary candidemia [263]. In sepsis, which has heterogeneous characteristics caused by various pathogens, IFN-γ can play dual roles in immune regulation. Therefore, its use in treatment should be tailored to the individual, guided by additional research.

#### 3.2.2. Interleukin-7 (IL-7)

Several studies have demonstrated a strong correlation between higher mortality and immunosuppression resulting from apoptosis-induced lymphocyte depletion in sepsis [264,265]. Interleukin-7 (IL-7) is a pluripotent cytokine that plays an important role in the survival and expansion of T cells and improves sepsis-induced lymphopenia survival [266,267]. The immunostimulatory effect of IL-7 has shown significant results in animal models of sepsis [268,269,270]. In addition, in an ex vivo study using the peripheral blood of patients with sepsis, IL-7 inhibited lymphocyte apoptosis and enhanced T cell cytokine production [187,271]. Based on the positive results of these preclinical studies, a phase 2 study was conducted in 2018 to evaluate the efficacy and safety of recombinant human IL-7 (rhIL-7, also known as CYT107). Profound sepsis-induced lymphopenia was successfully reversed through the intramuscular administration of CYT107, which resulted in a 3- to 4-fold increase in the total lymphocyte count and circulating CD4+ and CD8+ T cells. The effect of CYT107 on the maintenance of lymphocyte counts lasted for several weeks after the discontinuation of administration. However, there was no significant difference in survival at 28 and 120 days between the treatment and control groups [188]. A follow-up study that changed the injection route of CYT107 from intramuscular to intravenous was published in 2023. The improvement in sepsis-induced lymphopenia after the intravenous infusion of CYT107 was equivalent to that observed after intramuscular infusion. However, 3 of the 15 patients receiving intravenous CYT107 experienced fever and respiratory distress approximately 5–8 h after drug administration, leading to the early termination of this research [272]. Aside from the local side effects of intramuscular injection, the injection route of CYT107 seems to be reasonable for intramuscular rather than intravenous injection.

#### 3.2.3. Thymosin Alpha 1 (Tα1)

Tα1, a 28-amino acid peptide derived from the thymus, regulates both the innate and adaptive immune systems [273,274]. Tα1 acts as an agonist for TLR-2 and -9, enhancing the ability of antigen-presenting cells, such as dendritic cells, and promoting the maturation of T cells. Tα1 also plays a role in controlling inflammation by increasing the secretion of IL-10 and regulatory T cells [189,273]. Therefore, Tα1 may be a promising treatment option for sepsis with heterogeneous immune responses. In a multicenter RCT, including 361 patients with sepsis, the Tα1 group had a significantly lower hospital mortality rate than the control group. However, the 28-day mortality rate in the Tα1 group tended to be lower than that in the control group, but this was not statistically significant (*p* < 0.062). Furthermore, the Tα1 group showed a higher improvement in monocytic HLA-DR on days 3 and 7 compared to the control group, suggesting that Tα1 may enhance immune functions in sepsis [190]. In a subsequently published meta-analysis of 12 studies, including 1480 patients with sepsis, the Tα1 group had a survival benefit compared to the control group. However, caution is needed when interpreting the results because the included studies were of poor quality and the number of participants was small [275]. In 2016, a meta-analysis was published that evaluated whether a combined or single administration of ulinastatin and Tα1 affected the survival of patients with sepsis. In this study, combination treatment with ulinastatin and Tα1 reduced both the 28-day and 90-day mortality, but Tα1 monotherapy only reduced the 28-day mortality [276]. Finally, a large RCT (NCT02867267), including 1106 patients with sepsis, was recently completed to confirm the efficacy and safety of Tα1. Moreover, it is worth noting whether Tα1 was effective in treating sepsis in this study.

#### 3.2.4. Granulocyte-Macrophage Colony-Stimulating Factor (GM-CSF)

GM-CSF, a hematopoietic growth factor, stimulates the development of neutrophils, monocytes, macrophages, and dendritic cells and HLA-DR expression on monocytes [277,278]. Because of its ability to restore immune cell functions, GM-CSF has been widely used in studies on sepsis treatment [191,279]. An RCT conducted on 38 patients with sepsis-induced immunosuppression (monocytic HLA-DR < 8000 monoclonal antibodies per cell for 2 days) found that GM-CSF therapy was both effective and safe for restoring monocytic immunocompetence. Additionally, GM-CSF therapy reduces the amount of time that patients require mechanical ventilation and stays in the ICU or hospital [192]. In a meta-analysis published in 2015, GM-CSF treatment did not have a short-term survival benefit but improved clinical outcomes without fatal side effects. These clinical benefits include a more rapid recovery from infection, shorter hospital stays, fewer days requiring mechanical ventilation, and lower medical expenses [191]. The most recently published RCT focused on whether GM-CSF treatment could reduce secondary infections in immunosuppressed patients with sepsis (monocytic HLA-DR < 8000 antibodies bound per cell on day 3). This study was terminated early because of insufficient recruitment, and there was no significant difference in ICU-acquired infection (*p* < 1.000) or 28-day mortality (*p* < 0.900) between the GM-CSF and control groups [280]. A re-evaluation of the efficacy and safety of GM-CSF in patients with sepsis is needed in RCTs, including a more relaxed definition of immunosuppression and a larger number of patients.

#### 3.2.5. Mesenchymal Stem Cells (MSCs)

Mesenchymal stem cell (MSC) therapy for sepsis has recently gained attention owing to its immunomodulatory, anti-inflammatory, anti-apoptotic, and differentiation properties [281]. In addition, these cells have several advantages, including quick and easy isolation from various human tissues and relatively easy expansion [282]. The effectiveness of MSC treatment in sepsis is presumed to be due to immunomodulation, the repair of endothelial cell barrier damage, repair of tissue damage, and enhancement of bacterial clearance [283]. In several meta-analyses of animal models of sepsis, MSC treatment was shown to improve survival [284,285,286]. However, there were differences in the optimal MSC doses in each study. Therefore, large-scale studies are required to determine the optimal dose of MSCs for sepsis treatment. Based on the success of MSC treatment in animal studies, several trials have been conducted in patients with sepsis. Phase 1 clinical trials conducted on patients with sepsis confirmed the safety and tolerability of treatment with stem cells derived from the umbilical cord or bone marrow [287,288,289]. Phase 2 clinical trials are ongoing to determine the clinical efficacy of MSC therapy (NCT03369275, NCT05969275, NCT02883803, and NCT04961658). Based on these studies, it is necessary to confirm the potential of MSC therapy for the treatment of sepsis. At the same time, there is a lack of data on the safety of MSC treatment in sepsis compared to other diseases, such as acute respiratory distress syndrome; therefore, evaluation of this is necessary [290].

#### 3.2.6. Immune Checkpoint Inhibitors

Immune checkpoints are numerous inhibitory pathways in the immune system that are crucial for maintaining self-tolerance and regulating immune responses [291]. Several immune checkpoints have been identified, including the cytotoxic T-lymphocyte-associated protein 4 (CTLA-4), T-cell immunoglobulin and mucin domain-containing protein 3 (TIM-3), lymphocyte activation gene 3 (LAG-3), and V-domain Ig suppressor of T-cell activation (Figure 2). The most well-known immune checkpoint in sepsis is the programmed cell death protein 1 (PD-1) pathway [292]. This system consists of the receptor PD-1 and its ligands, PD-L1 and PD-L2, which play a role in regulating the activation of T cells. In sepsis, the overexpression of PD-1 and its ligands, PD-L1 and PD-L2, is associated with decreased cytokine production, defective antigen presentation, impaired humoral immunity, and decreased phagocytosis [293,294,295]. In a postmortem analysis of patients with sepsis, the expression of PD-1 in splenic T cells, PD-L1 in splenic capillary endothelial cells, and lung tissues was higher than that in the control group [296]. In a murine model of experimental sepsis, PD-1−/− mice exhibited decreased organ damage, lower cytokine levels, and a reduced bacterial burden. They are less vulnerable to cecal ligation and puncture-induced lethality than wild-type mice [297]. Furthermore, ex vivo studies on cells from patients with sepsis have revealed that inhibition of the PD-1/PD-L1 pathway reduces apoptosis, improves immune cell functions, and increases cytokine production [193,294]. The anti-PD-L1 monoclonal antibody was well tolerated in the first clinical trial of PD-1/PD-L1 pathway blockade in sepsis, and at higher dosages, it could restore the immunological state [194]. Other phase 1 clinical trials have shown that nivolumab, an anti-PD-1 antibody, is well tolerated in the treatment of patients with sepsis or septic shock. Additionally, nivolumab improves immune functions by increasing monocytic HLA-DR expression and absolute lymphocyte counts [298,299]. Strategies that modulate immune checkpoints in the treatment of sepsis seem to be promising, but caution is required because the blockade of PD-1/PD-L1 can trigger the immune system to attack healthy cells, resulting in various immune-related adverse events [300].

## 4. Conclusions and Future Direction

Sepsis remains a leading cause of death worldwide, highlighting the urgent need for more accurate and rapid diagnostic strategies. While numerous studies have explored the use of biomarkers for sepsis diagnosis, single biomarkers often fall short in sensitivity and specificity. Combining biomarkers has shown promise in improving the diagnostic accuracy, though this approach requires further clinical validation [301,302]. Efforts have been made to integrate clinical information into biomarker panels [303,304,305], and recent approaches include machine learning models combined with biomarkers, genomics, and electronic medical record data for the early diagnose of sepsis [306,307,308]. For example, Wang et al. identified three genes (*COMMD9*, *CSF3R*, and *NUB1*) as potential biomarkers related to immune cell infiltration and sepsis prediction [309]. Another study identified eight genes associated with sepsis severity and prognosis [310]. Machine learning has also shown potential in predicting sepsis outcomes. The eXtreme Gradient Boosting model has demonstrated high performance in predicting in-hospital death among patients with sepsis, with key determinants, including biomarkers, advanced age, and clinical indicators [311]. The future diagnosis of sepsis lies in the convergence of clinical assessments, biomarkers, and intelligent bioinformatics tools, aiming for early identification and improved patient outcomes.

Despite significant technological and medical advancements, a universally effective treatment for sepsis remains elusive due to several interrelated and complex factors. The inherent complexity and heterogeneity of sepsis, triggered by various pathogens each eliciting different immune responses, complicate the development of one-size-fits-all therapies. Patient variability in genetics, health conditions, age, and immune status further complicates treatment. Furthermore, sepsis is characterized by a dynamic and dysregulated immune response, involving an initial hyperinflammatory phase that can transition to a state of immunosuppression. This unpredictable and fluctuating progression makes it challenging to time treatments effectively. The immune system’s dysregulation during sepsis, where simultaneous hyperinflammation and immunosuppression can occur in different parts of the body or at different times, adds another layer of complexity. To overcome these challenges and develop effective treatments for sepsis, the field of personalized medicine holds promise, with the potential to tailor treatments to the individual’s genetic makeup, immune status, and the specific pathogen involved. In this context, stratifying patients using biomarkers or omics-based technology and administering immunotherapies according to sepsis endotypes has gained attention [312,313]. Comprehensive care strategies that integrate early recognition, rapid intervention, and continuous monitoring to adapt treatment plans in real time are necessary to address the dynamic nature of sepsis. 

## Figures and Tables

**Figure 1 ijms-25-07396-f001:**
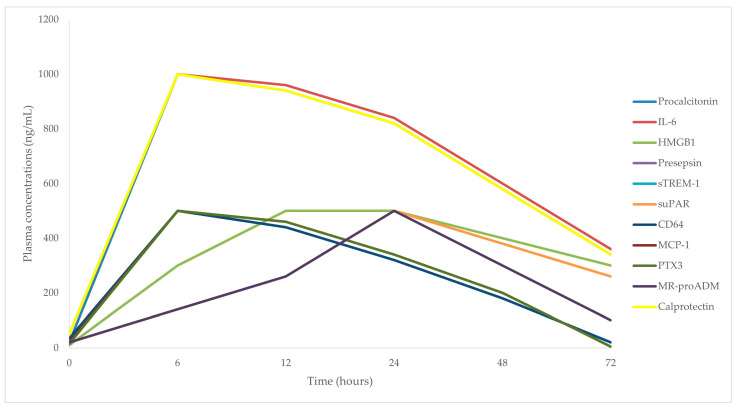
Temporal dynamics of plasma biomarker concentrations in sepsis. IL-6, interleukin-6; HMGB1, high mobility group box 1; sTREM-1, soluble triggering receptor expressed on myeloid cell-1; suPAR, soluble urokinase plasminogen activator receptor; MCP-1, monocyte chemoattractant protein-1; PTX3, pentraxin 3; MR-proADM, mid-regional proadrenomedullin.

**Figure 2 ijms-25-07396-f002:**
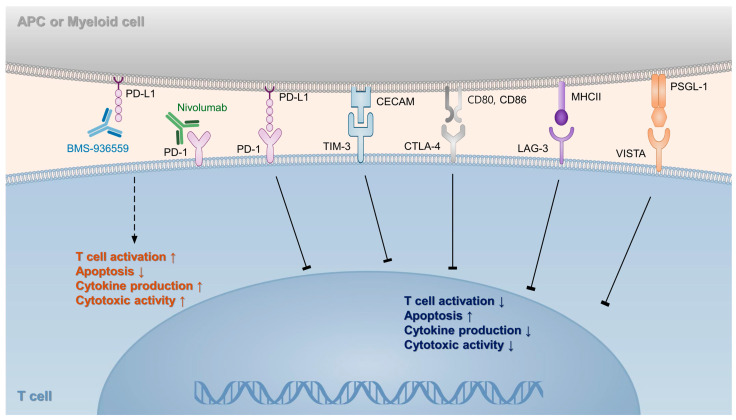
Overview of immune checkpoint pathways in sepsis. Nivolumab and BMS-936559 prevent T cell exhaustion during sepsis by inhibiting the binding of PD-1 and PD-L1. PD-1, programmed death-1; PD-L1, programmed death ligand-1; TIM-3, T cell membrane protein-3; CECAM, carcinoembryonic antigen-related cell adhesion molecule; CTLA-4, cytotoxic T-lymphocyte-associated protein 4; LAG-3, lymphocyte-activation gene 3; MHC II, major histocompatibility complex II; VISTA, V-domain Ig suppressor of T cell activation; PSGL-1, P-selectin glycoprotein ligand-1.

**Table 1 ijms-25-07396-t001:** Overview of biomarkers for diagnosing sepsis based on pathophysiology.

Category	Biomarker	Demographic	Variables	AUC/95% CI	Sensitivity/Specificity/PLR/NLR/DOR	Clinical Relevance	Refs.
**APPs**	PCT	Meta-analysis: 22 studies; 2680	PCT	0.83	Sensitivity: 0.78 (0.75–0.80); Specificity: 0.67 (0.64–0.70); PLR: 2.68 (2.18–3.28); NLR: 0.27 (0.20–0.36); DOR: 11.61 (7.04–19.15)	Differentiating sepsis from non-infectious SIRS	[27]
	PCT	Meta-analysis: 39 studies	PCT	0.87 (0.83–0.89)	Sensitivity: 0.82 (0.78–0.85); Specificity: 0.78 (0.74–0.82); PLR: 3.7 (3.1–4.50); NLR: 0.23 (0.19–0.29); DOR: 16 (11–23)		[28]
	PCT	Meta-analysis: 19 studies; 3012 patients	PCT	0.84 (0.81–0.87)	Sensitivity: 0.80 (0.75–0.84); Specificity: 0.75 (0.67–0.81)		[29]
	PTX3	Prospective study; sepsis:51; septic shock: 46; control: 45	PTX3	0.84 (0.78–0.91)	Sensitivity: 76.3%; Specificity: 80.0%	Diagnostic value: IL6 > PTX3	[30]
**DAMPs**	Calprotectin	Meta-analysis, 6 studies; 821 patients	Calprotectin	0.88	Sensitivity: 0.77 (0.62–0.87); Specificity: 0.85 (0.74–0.92); PLR: 5.20 (2.75–9.84); NLR: 0.27 (0.15–0.48); DOR: 19.37 (6.71–55.92)		[31]
	HMGB1	SIAKI: 50 N-AKI: 70	HMGB1 (blood + urine)	0.891	Sensitivity: 0.88; Specificity: 0.87; Accuracy: 0.88	Diagnosis of SIAKI	[32]
	HMGB1	Sepsis: 66 Control: 78	HMGB1	0.684	Sensitivity: 75.8%; Specificity: 41.3%	Diagnosis of sepsis	[33]
			HBD-3	0.679	Sensitivity: 63.6%; Specificity: 93.5%		
**Membrane receptors**	nCD64	Meta-analysis, 8 studies; 1986 patients	CD64	0.95 (SROC)	Sensitivity: 0.76 (0.73–0.78); Specificity: 0.85 (0.82–0.87); PLR: 8.15 (3.82–17.36); NLR: 0.16 (0.09–0.30); DOR: 60.41 (15.87–229.90)		[34]
	nCD64	Meta-analysis, 14 studies; total 2471 patients	CD64	SROC: 0.94 (0.80–0.92)	Sensitivity: 0.87 (0.80–0.92); Specificity: 0.89 (0.82–0.93); PLR: 7.8 (4.7–13.1); NLR: 0.15 (0.09–0.25); DOR: 53 (22–128)	In diagnosis of sepsis: CD64 > PCT or CRP	[35]
		927 patients	CD64 vs. PCT	0.89 vs. 0.84			
		744 patients	CD64 vs. CRP	0.89 vs. 0.84			
	nCD64	Meta-analysis: 54 studies	CD64	0.94 (0.91–0.96)	Sensitivity: 0.88 (0.81–0.92); Specificity: 0.88 (0.83–0.91); PLR: 7.2 (5.0–10.3); NLR: 0.14 (0.09–0.22); DOR: 51 (25–105)		[28]
**Soluble receptors**	Presepsin	Meta-analysis: 9 studies; Sepsis: 1320; SIRS (non-infection): 512; Healthy control: 327	Presepsin	0.89 (0.84–0.94)	Sensitivity: 0.78 (0.76–0.80); Specificity: 0.83 (0.80–0.85); PLR: 4.63 (3.27–6.55); NLR: 0.22 (0.16–0.30); DOR: 21.73 (12.81–36.86)		[36]
	Presepsin	Meta-analysis: 11 studies; Sepsis: 1630; Control: 1422	Presepsin	0.88 (0.84–0.90)	Sensitivity: 0.83 (0.77–0.88); Specificity: 0.78 (0.72–0.83)		[37]
	Presepsin	Meta-analysis: 8 studies; Sepsis: 1165; SIRS (non-infection): 525	Presepsin	0.89 (0.86–0.92)	Sensitivity: 0.86 (0.79–0.91); Specificity: 0.78 (0.68–0.85); PLR: 3.8 (2.6–5.7); NLR: 0.18 (0.11–0.28); DOR: 22 (10–48)		[38]
	Presepsin	Meta-analysis: 19 studies; 3012 patients	Presepsin	0.87 (0.84–0.90)	Sensitivity: 0.84 (0.80–0.88); Specificity: 0.73 (0.61–0.82)		[29]
	sTREM-1	Meta-analysis: 19 studies; 2418 patients	sTREM-1	0.88 (0.85–0.91)	Sensitivity: 0.82 (0.73–0.89); Specificity: 0.81 (0.75–0.86); PLR: 4.3 (3.02–6.12); NLR: 0.22 (0.24–0.35); DOR: 20 (9–41)		[39]
	sTREM-1	Meta-analysis: 21 studies; 2401 patients	sTREM-1	0.89 (0.85–0.91)	Sensitivity: 0.85 (0.76–0.91); Specificity: 0.79 (0.70–0.86); PLR: 4.0 (2.7–6.0); NLR: 0.19 (0.11–0.33); DOR: 21 (9–49)		[40]
	suPAR	Meta-analysis: 7 studies; Sepsis: 1062	suPAR	0.82 (0.78–0.85)	Sensitivity: 0.67 (0.53–0.79); Specificity: 0.80 (0.72–0.86); PLR: 3.4 (2.1–5.3); NLR: 0.41 (0.26–0.65); DOR: 8 (3–20)		[41]
	suPAR	Meta-analysis: 17 studies; 2722	suPAR	0.83 (0.80–0.86)	Sensitivity: 0.76 (0.63–0.86); Specificity: 0.78 (0.72–0.83); PLR: 3.50 (2.60–4.70); NLR: 0.30 (0.18–0.50); DOR: 12 (6–24)		[42]
**Cytokines and chemokines**	IL-6	Meta-analysis: 22 studies; 2680	IL-6	0.80	Sensitivity: 0.68 (0.65–0.70); Specificity: 0.73 (0.71–0.76); PLR: 2.46 (1.96–3.08); NLR: 0.42 (0.33–0.53); DOR: 7.05 (4.48–11.10)	IL-6 showed diagnostic value comparable to PCT	[27]
	IL-6	Meta-analysis: 15 studies	IL-6	0.77 (0.73–0.80)	Sensitivity: 0.72 (0.65–0.78); Specificity: 0.70 (0.62–0.76); PLR: 2.4 (1.9–3.0); NLR: 0.4 (0.32–0.51); DOR: 6 (4.0–9.0)		[28]
	IL-6	Prospective study; sepsis:51; septic shock: 46; control: 45	IL-6	0.89 (0.83–0.94)	Sensitivity: 80.4% Specificity: 88.9%	Diagnostic value: IL6 > PTX3	[30]
	MCP-1	Meta-analysis: 8 studies; 805	MCP-1	0.90 (0.87–0.92)	Sensitivity: 0.84 (0.70–0.92); Specificity: 0.82 (0.67–0.91); PLR: 3.71 (2.12–6.50); NLR: 0.287 (0.20–0.42); DOR: 16.508 (7.63–35.71)		[43]
**Endothelial dysfunction markers**	MR-proADM	Meta-analysis: 40 studies	MR-proADM	0.91 (0.88–0.93)	Sensitivity: 0.84 (0.78–0.88); Specificity: 0.86 (0.79–0.91); PLR: 5.8 (3.8–9.0); NLR: 0.19 (0.14–0.27); DOR: 31 (15–62)	Diagnostic accuracy: MR-proADM > Presepsin	[44]
			Presepsin	0.90 (0.87–0.92)	Sensitivity: 0.86 (0.82–0.90); Specificity: 0.79 (0.71–0.85); PLR: 4.0 (3.0–5.5); NLR: 0.18 (0.13–0.23); DOR: 23 (14–36)		
	Ang	Prospective study; severe sepsis: 105	Ang2	0.97	Sensitivity: 0.9 Specificity: 0.99	Diagnostic accuracy: Ang2 > Ang1	[45]
			Ang1	0.66	Sensitivity: 0.63 Specificity: 0.65		
			Ang1/Ang2	0.66	Sensitivity: 0.93 Specificity: 0.46		
**Organ dysfunction markers**	miRNAs	Meta-analysis: 30 studies; 3914 patients	miRNAs	0.89 (0.86–0.92)	Sensitivity: 0.80 (0.75–0.83); Specificity: 0.85 (0.80–0.89); PLR: 5.3 (4.0–6.9); NLR: 0.24 (0.20–0.29); DOR: 22 (15–32)	Diagnostic accuracy: miRNAs > PCT, CRP	[46]
		6 studies; 732 patients	miR-223	0.87 (0.84–0.90)	Sensitivity: 0.77 (0.67–0.84); Specificity: 0.91 (0.73–0.97); PLR: 8.3 (2.5–27.9.); NLR: 0.25 (0.17–0.38); DOR: 33 (8–142)		

AUC, area under the curve; CI, confidence interval; PLR, positive likelihood ratio; NPR, negative likelihood ratio; DOR, diagnostic odds ratio; Ref, reference; APP, acute phase protein; PCT, procalcitonin; IL-6, interleukin-6; PTX3, pentraxin 3; DAMPs, damage associated molecular patterns; HMGB1, high mobility group box 1; SIAKI, sepsis-associated acute kidney injury; N-AKI, no-acute kidney injury; HBD-3, human β-defensin 3; SROC, summary receiver operating characteristics; SIRS, systemic inflammatory response syndrome; sTREM-1, soluble triggering receptor expressed on myeloid cell-1; suPAR, soluble urokinase plasminogen activator receptor; CRP, C-reactive protein; nCD64, neutrophil CD64; MCP-1, monocyte chemoattractant protein-1; MR-proADM, mid-regional proadrenomedullin, Ang, angiopoietin; miRNA, microRNA.

**Table 2 ijms-25-07396-t002:** Overview of biomarkers for differentiating sepsis from SIRS.

Category	Biomarker	Cut-Off Value	Demographic	AUC/95% CI	Sensitivity/Specificity/PLR/NLR/DOR	Refs.
**APPs**	PCT	0.96 ng/mL	Meta-analysis: 59 studies; 7376	0.85 (0.82–0.88)	Sensitivity: 0.79 (0.75–0.83) Specificity: 0.78 (0.74–0.81)	[47]
	PCT	2.2 ng/mL	Septic shock: 24; Severe sepsis: 31; SIRS: 11	0.801	Sensitivity: 56.4%; Specificity: 100%; PPV: 100%; NPV: 31.4%	[48]
	PCT	1.57 ng/mL	Sepsis: 52; SIRS: 38	0.65	Sensitivity: 67.31%; Specificity: 65.79%; PPV: 72.92; NPV: 59.52	[49]
	PCT		Meta-analysis: 21 studies; 2620	0.83	Sensitivity: 0.78 (0.75–0.80); Specificity: 0.67 (0.64–0.70); PLR: 2.68 (2.18–3.28); NLR: 0.27 (0.20–0.36); DOR: 11.61 (7.04–19.15)	[27]
	CRP	84 mg/L	Meta-analysis: 45 studies; 5654	0.77 (0.73–0.81)	Sensitivity: 0.75 (0.69–0.79); Specificity: 0.67 (0.58–0.74)	[47]
	CRP		Sepsis: 72; SIRS (nonbacterial): 23	0.859		[50]
**Membrane receptors**	CD64		Meta-analysis: 4 studies; 558	0.996 (0.94–0.97)	Sensitivity: 0.87 (0.75–0.94); Specificity: 0.93 (0.87–0.96)	[47]
	CD64	4300 molecular per neutropthil	Septic shock: 24; Severe sepsis: 31; SIRS: 11	0.928	Sensitivity: 89.1%; Specificity: 95.9%; PPV: 98%; NPV: 62.5%	[48]
	CD64		Septic shock: 55; Severe sepsis: 34; Sepsis: 59; SIRS (nonbacterial): 145	0.80 (0.75–0.84)	Sensitivity: 63%; Specificity: 89%; PPV: 85.3%; NPV: 70.1%	[51]
**Soluble receptors**	Presepsin	600 pg/mL	Meta-analysis: 9 studies; 1510	0.88 (0.85–0.90)	Sensitivity: 0.84 (0.79–0.88); Specificity: 0.77 (0.68–0.84)	[47]
	Presepsin	470 pg/mL	Sepsis: 72; SIRS (nonbacterial): 23	0.954	Sensitivity: 98.6%; Specificity: 82.6%	[50]
	sTREM-1	123 pg/mL	Meta-analysis: 8 studies; 831	0.85 (0.82–0.88)	Sensitivity: 0.78 (0.66–0.87); Specificity: 0.78 (0.65–0.87)	[47]
	sTREM-1	133 pg/mL	Sepsis: 52; SIRS: 38	0.78	Sensitivity: 71.15%; Specificity: 76.32%; PPV: 80.43; NPV: 65.91	[49]
	sTREM-1	49 pg/mL	Sepsis: 42; SIRS: 25	1.0 (First day) 0.93 (Seventh day)	Sensitivity: 100%; Specificity: 84%	[52]
	sTREM-1	30–60,000 pg/mL	Meta-analysis: 21 studies; 2401 patients	0.89 (0.85–0.91)	Sensitivity: 0.85 (0.76–0.91); Specificity: 0.79 (0.70–0.86); PLR: 4.0 (2.7–6.0); NLR: 0.19 (0.11–0.33); DOR: 21 (9–49)	[40]
	suPAR	6.4 ng/mL	Meta-analysis: 4 studies; 481	0.68 (0.64–0.72)	Sensitivity: 0.61 (0.53–0.68); Specificity: 0.82 (0.63–0.93); PLR: 3.4 (1.4–8.5); NLR: 0.48 (0.34–0.67); DOR: 7 (2–25)	[41]
	suPAR	7.5 ng/mL	Meta-analysis: 5 studies; 637	0.81 (0.77–0.84)	Sensitivity: 0.67 (0.58–0.76); Specificity: 0.82 (0.73–0.88); PLR: 3.70 (2.40–5.80); NLR: 0.40 (0.30–0.53); DOR: 9 (5–18)	[42]
**Cytokines and chemokines**	IL-6	138 pg/mL	Meta-analysis: 22 studies; 3450	0.79 (0.75–0.82)	Sensitivity: 0.72 (0.63–0.80) Specificity: 0.73 (0.67–0.79)	[47]
	IL-6	18–423.5 pg/mL	Meta-analysis: 22 studies; 2680	0.80	Sensitivity: 0.68 (0.65–0.70); Specificity: 0.73 (0.71–0.76); PLR: 2.46 (1.96–3.08); NLR: 0.42 (0.33–0.53); DOR: 7.05 (4.48–11.10)	[53]

AUC, area under the curve; CI, confidence interval; PPV, positive predictive value; NPV, negative predictive value; Ref, reference; APP, acute phase protein; PCT, procalcitonin; SIRS, systemic inflammatory response syndrome; PLR, positive likelihood ratio; NPR, negative likelihood ratio; DOR, diagnostic odds ratio; CRP, C-reactive protein; sTREM-1, soluble triggering receptor expressed on myeloid cell-1; suPAR, soluble urokinase plasminogen activator receptor; IL-6, interleukin-6.

**Table 3 ijms-25-07396-t003:** Overview of immunomodulation therapies in sepsis.

Immunomodulating Agents	Types of Study	Study Population	Primary Endpoint	Main Results	Refs.
**rhIL-1ra**	mRCT	Sepsis (*n* = 893)	28-day all-cause mortality	No survival benefit compared to placebo.	[169]
**rhIL-1ra**	Phase III mRCT	Severe sepsis or septic shock (*n* = 696)	Efficacy and safety	Terminated prematurely as no significant survival benefit was observed.	[170]
**rhIL-1ra**	Reanalysis of phase III mRCT	Severe sepsis or septic shock (*n* = 763)	28-day survival	The use of rhIL-1ra in the sepsis with HBD/DIC showed improved survival.	[171]
**IL6RA**	Meta-analysis	Sepsis (*n* = 11,643)	Incidence of sepsis, 28-day mortality	Reduced incidence of sepsis. Improved survival even in non-COVID-19 sepsis.	[172]
**Vilobelimab (recombinant monoclonal anti-C5a Ab)**	Phase II mRCT	Severe sepsis or septic shock (*n* = 72)	Safety, tolerability, pharmacokinetics, and pharmacodynamics of vilobelimab	C5a decreased in a dose-dependent manner. Tolerable and safe. ICU- and ventilator-free days ↑. No survival benefits.	[173]
**Adrecizumab** **(non-neutralizing monoclonal anti-ADM Ab)**	Phase IIa mRCT	Septic shock within 12 h of vasopressor start and ADM > 70 pg/mL (*n* = 301)	Safety and tolerability	Tolerable and safe. No difference in 28-day mortality (Adrecizumab 23.9% vs. placebo 27.7%).	[174]
**Adrecizumab**	Phase II mRCT	Septic shock within 12 h of vasopressor start and ADM > 70 pg/mL (*n* = 301)	cDPP3-based enrichment on treatment efficacy of Adrecizumab	In subgroup with cDDP3 <50 ng/mL, HR for 28-day mortality tended to improve.	[175]
**Nangibotide**	Phase IIa mRCT	Septic shock (*n* = 49)	Safety and tolerability, pharmacodynamics, pharmacokinetics	Tolerable and safe. Improvement in SOFA score.	[176]
**Nangibotide**	Phase IIb mRCT	Septic shock (*n* = 355)	Mean differences in total SOFA score from baseline to day 5	Did not achieve primary outcome. High concentration of nangibotide led to improvement in SOFA score (when the levels of predefined sTREM-1 > 532 pg/mL).	[177]
**Polymyxin B hemoperfusion**	mRCT	Severe sepsis or septic shock (*n* = 64)	Change in MAP and vasopressor requirement	Increased MAP. Decreased vasopressor requirement. Reduced 28-day mortality.	[178]
**Polymyxin B hemoperfusion**	mRCT	Septic shock due to peritonitis (*n* = 243)	28-day mortality	No survival benefits. No improvement in organ failure.	[179]
**Polymyxin B hemoperfusion**	mRCT	Septic shock and EAA ≥ 0.60 (*n* = 450)	28-day mortality	No survival benefits.	[180]
**Polymyxin B hemoperfusion**	mRCT	Septic shock and EAA ≥ 0.6–0.89 (*n* = 194)	28-day mortality (adjusted for APACHE II score and baseline MAP)	Reduced 28-day mortality. Increased MAP and VFDs.	[181]
**CytoSorb^®^**	mRCT	Sepsis or septic shock on MV (*n* = 97)	Changes in IL-6 concentration between day 1 and 7	No effect on changes in IL-6 concentration.	[182]
**CytoSorb^®^**	Meta-analysis (6 studies)	Sepsis or septic shock (*n* = 413)	Mortality at 28–30 days	No survival benefits.	[183]
**oXiris^®^**	Randomized crossover double-blind study	Septic shock with stage 3 ARF and plasma endotoxic levels > 0.03 EU/mL (*n* = 16)	Changes in plasma endotoxin level	Reduced endotoxin. Reduced cytokines (TNF-α, IL-6, IL-8, INF-γ).	[184]
**oXiris^®^**	Meta-analysis (14 studies)	Sepsis undergoing CRRT	28-day mortality	Reduced 28-, 7-, and 14- day mortalities. Decrease in SOFA score, lactate levels, NE dose.	[185]
**IFN-γ**	Phase II mRCT	Sepsis with signs of immunoparalysis	Difference in the mean total SOFA score until day 9 after randomization	In progress.	[186]
**Intramuscular rhIL-7**	Phase IIb mRCT	Septic shock (*n* = 27)	Safety and ability to reverse lymphopenia	Tolerable and safe. 3–4 fold increase in ALC. No survival benefits.	[187]
**Intravenous rhIL-7**	Phase IIb mRCT	Septic shock and ALCs ≤ 900 cells/mm^3^ (*n* = 40)	Changes in ALC at day 29	2–3 fold increase in ALC. This study was terminated early due to transient respiratory distress.	[188]
**Tα1**	mRCT	Severe sepsis (*n* = 361)	28-day all-cause mortality	Reduced 28-day mortality but marginal *p*-value (*p* = 0.049). Increased expression of mHLA-DR.	[189]
**Tα1**	Meta-analysis (12 studies)	Sepsis (*n* = 1480)	All-cause mortality	Reduced all-cause mortality.	[190]
**Tα1**	Phase III mRCT	Sepsis (*n* = 1106)	28-day mortality	Pending (NCT02867267).	
**GM-CSF**	mRCT	Severe sepsis or septic shock and reduced levels of mHLA-DR (*n* = 38)	mHLA-DR expression	Increased expression of mHLA-DR. VFD ↑ Length of hospital or ICU stay ↓.	[191]
**GM-CSF**	mRCT	Severe sepsis or septic shock and reduced levels of mHLA-DR (*n* = 98)	ICU-acquired infection at day 28 or ICU discharge	Not effective in reducing ICU-acquired infections. Terminated early due to insufficient recruitment.	[192]
**MSC**	Phase II mRCT	Septic shock (*n* = 114)	Efficacy and safety	Pending (NCT03369275).	
**MSC**	Phase II mRCT	Septic shock (*n* = 296)	Ventilator-, vasopressor-, RRT-free days	Pending (NCT05969275).	
**Anti-PD-L1 Ab**	Phase Ib mRCT	Sepsis and ALC ≤ 1100 cells/μL	Death and AEs	Tolerable and safe. Restored mHLA-DR expression at higher doses.	[193]
**Nivolumab** **(anti-PD-1 Ab)**	Phase Ib mRCT	Sepsis and ALC ≤ 1100 cells/μL	Safety, tolerability, pharmacokinetics	Tolerable and safe. Increased levels of mHLA-DR.	[194]

rhIL-1ra, recombinant human interleukin-1 receptor antagonist; mRCT, multicenter randomized controlled trial; HBD, hepatobiliary dysfunction; DIC, disseminated intravascular coagulation; IL6RA, interleukin-6 receptor antagonist; COVID-19, coronavirus disease 2019; Ab, antibody; ADM, adrenomedullin; cDDP3, circulating dipeptidyl peptidase 3; HR, hazard ratio; SOFA, sequential organ failure assessment; sTREM, soluble triggering receptor expressed on myeloid cell-1; MAP, mean arterial pressure; EAA, endotoxin activity assay; APACHE II, acute physiologic and chronic health evaluation II; VFD, ventilator-free days; MV, mechanical ventilation; ARF; acute renal failure; TNF-α, tumor necrosis factor-α; IFN-γ, interferon-γ; CRRT; continuous renal replacement therapy; NE, norepinephrine; rhIL-7, recombinant human interleukin-7; ALC, absolute lymphocyte count; Tα1, thymosin alpha 1; mHLA-DR, monocytic human leukocyte antigen-DR; GM-CSF, granulocyte-macrophage colony-stimulating factor; MCS, mesenchymal stem cells; RRT, renal replacement therapy; AEs, adverse events.

## Data Availability

Data sharing is not applicable to this article.

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
