# Peer review of "Navigating the Modern Landscape of Sepsis: Advances in Diagnosis and Treatment"

_ijms, 2024, doi:10.3390/ijms25137396_

Round 1

Reviewer 1 Report

Comments and Suggestions for Authors

The manuscript of Jang et al. reviewed progress in the research of sepsis biomarkers and therapeutic approaches. Early diagnosis and prompt treatment are essential for sepsis patients. But it is challenging to define and early recognize sepsis, so combining biomarkers and clinical data is potentially useful for clinician to improve timeliness and accuracy of diagnosis. The reviewer has the following comments:

  1. The authors state: “the sequental organ failure assessment (SOFA) score has been used as an index to predict organ failure, but there is a limitation that it is not specific for sepsis or infection”. Please comment on Quick Sepsis-related Organ Failure Assessment (qSOFA) and National Early Warning Score (NEWS). And compared to biomarkers, what are the different roles of these methods in the practice of rapid diagnosis of sepsis?

2. The authors report that “sTREM-1 showed the ability to differentiate between sepsis and SIRS”. The reviewer suggests adding a figure or table listing biomarkers assumed to be able to differentiate infectious and non-infectious causes of critical illness. This will be of great benefit to the readers.

3. The authors suggest that “machine learning models” are a valid method to combine biomarkers, genomics and electronic medical record data to diagnose sepsis early, but more information about machine learning models, and the important role in improving the survival of patients with sepsis should be provided. For which biomarkers machine learning models have shown prognostic value?

4. Immunomodulatory therapies play potentially an important role in improving the survival of patients with sepsis. The reviewer recommends to expand on the dynamic immune status of sepsis patients ranging from hyperinflammation to immunosuppression.

5. The conclusion should be concise, showing the most important and useful results or suggestions to the reader.

Reviewer 2 Report

Comments and Suggestions for Authors

Good review.

The authors need to discuss why despite so many advances in the area of sepsis still there is no adequate treatment. What are the lacunae of the current understanding and why need to be done in future to understand about sepsis. What new treatments or approaches to treatment are suggestions of the authors in this regard.

Comments on the Quality of English Language

ok

Reviewer 3 Report

Comments and Suggestions for Authors

Sepsis is a global health concern with high morbidity and mortality rates. Traditional diagnostic methods are time-consuming and contribute to high mortality. Biomarkers have been developed to diagnose, predict, and assess sepsis. Over 250 biomarkers have been identified, but no specific treatment is available. Selecting the most suitable patient is crucial for sepsis-specific treatment. Certain issues require resolution.

1.     Table 1 provides an overview of the biomarkers associated with sepsis diagnosis, organized based on their pathophysiology. Nevertheless, grasping the connection between them can be quite difficult.

2.     Readers may have concerns regarding the speed of responsiveness in the sepsis process. The authors offer valuable insights into the timing of appearance in patients with sepsis.

3.     It would be helpful if the authors could include the curves depicting the changes in biomarkers during sepsis. The reader will find it easier to understand.

4.     Concerning “3.1: Targeting the hyperinflammation of sepsis”, the readers would highly value it if the authors could consolidate the information from the existing research into a concise and comprehensible message.

5.     On page 21, the authors describe immune checkpoints, which are important inhibitory pathways in the immune system. These pathways play a critical role in maintaining self-tolerance and regulating immune responses. The presented material is distinctive and would be improved by using figures to enhance clarity.

Provided that the authors include more informative figures that depict the mechanism of action in the biomarkers, I am confident that the manuscript will be skillfully written and worthy of being published.

Comments on the Quality of English Language

The English language may be improved with some moderate editing.

Round 2

Reviewer 3 Report

Comments and Suggestions for Authors

This manuscript has undergone significant revisions by the authors. The biomarkers of sepsis have been reclassified and unnecessary details have been removed from the table to enhance its readability. The authors have provided visual representations of the changes in biomarker concentrations over time in sepsis (Figure 1) and immune checkpoint pathways in sepsis (Figure 2). These figures effectively address the concerns of the readers.

Comments on the Quality of English Language

The English language proficiency is satisfactory.

Author Response

Comments 1:

This manuscript has undergone significant revisions by the authors. The biomarkers of sepsis have been reclassified and unnecessary details have been removed from the table to enhance its readability. The authors have provided visual representations of the changes in biomarker concentrations over time in sepsis (Figure 1) and immune checkpoint pathways in sepsis (Figure 2). These figures effectively address the concerns of the readers.

Response 1: 

Thank you for your detailed and thoughtful feedback on our manuscript. We greatly appreciate your recognition of the significant revisions we have made. We are pleased to hear that the reclassification of sepsis biomarkers and the removal of unnecessary details from the table have enhanced its readability. We are also glad that the visual representations of biomarker concentration changes over time (Figure 1) and immune checkpoint pathways in sepsis (Figure 2) have effectively addressed the concerns of the readers. Your constructive comments have been invaluable in improving the quality and clarity of our work.

Thank you once again for your time and effort in reviewing our manuscript.